# Preventing PTSD, depression and associated health problems in student paramedics: protocol for PREVENT-PTSD, a randomised controlled trial of supported online cognitive training for resilience versus alternative online training and standard practice

Jennifer Wild,[1,2] Shama El-Salahi,[1] Gabriella Tyson,[1] Hjördis Lorenz,[1] Carmine M Pariante,[3] Andrea Danese,[4,5] Apostolos Tsiachristas,[6] Edward Watkins,[7] Benita Middleton,[8] Amanda Blaber,[9] Anke Ehlers[1,2]

For numbered affiliations see end of article.

**Correspondence to**
Dr Jennifer Wild;
jennifer.wild@psy.ox.ac.uk

## ABSTRACT

**Introduction** Emergency workers dedicate their lives to promoting public health and safety, yet suffer higher rates of post-traumatic stress disorder (PTSD) and major depression (MD) compared with the general population. They also suffer an associated increased risk for physical health problems, which may be linked to specific immunological and endocrine markers or changes in relevant markers. Poor physical and mental health is costly to organisations, the National Health Service and society. Existing interventions aimed at reducing risk of mental ill health in this population are not very successful. More effective preventative interventions are urgently needed. We first conducted a large-scale prospective study of newly recruited student paramedics, identifying two cognitive factors (rumination and resilience appraisals) that predicted episodes of PTSD and MD over a 2-year period. We then developed internet-delivered cognitive training for resilience (iCT-R), a supported online intervention, to modify cognitive predictors. This protocol is for a randomised controlled trial to evaluate the efficacy of the resilience intervention.

**Methods and analysis** 570 student paramedics will be recruited from participating universities. They will be randomly allocated to iCT-R or to supported online training of an alternative, widely available intervention or to training-as-usual. Follow-up will occur after the intervention/standard practice period and at 6, 12 and 24 months. Primary outcomes include rates of PTSD and MD and subsyndromal PTSD and MD, measured by the Structured Clinical Interview for Diagnostic and Statistical Manual of Mental Disorders, fifth edition, the Patient-Health Questionnaire-9 and the Post-traumatic Stress Disorder Checklist for Diagnostic and Statistical Manual of Mental Disorders, fifth edition. Secondary outcomes include measures of resilience, rumination, anxiety, psychological distress, well-being, salivary cortisol, plasma levels of C-reactive protein, smoking and alcohol use,

## Strengths and limitations of this study

- ► The study is a large, single-blind randomised controlled trial of internet-delivered cognitive training for resilience (iCT-R).
- ► iCT-R will be evaluated in comparison to an existing intervention and treatment as usual.
- ► Primary outcomes will be assessed by self-report and structured interviews.
- ► Full outcome blinding is not possible.
- ► Smoking and alcohol use will be measured with unpublished self-report tools.

weight gain, sleep problems, health-related quality of life, health resource utilisation and productivity.

**Ethics and dissemination** The Medical Sciences Inter-Divisional Research Ethics Committee at the University of Oxford granted approval, reference: R44116/RE001. The results will be published in a peer-reviewed journal. Access to raw data and participant information will be available only to members of the research team.

**Trial registration number** ISRCTN16493616; Pre-results.

## INTRODUCTION

Emergency workers carry a threefold increase, compared with the general population, in risk for major depression (MD) and post-traumatic stress disorder (PTSD) and an associated increased risk of poor physical health.[1] To date, interventions aimed at reducing risk of ill mental health in this population have been unsuccessful. Randomised controlled trials (RCT) found that trauma risk management, a peer support system widely available

to the police and ambulance services in England[2]; critical incident stress debriefing widely used by UK fire-services,[3] and the charity Mind's six-session group-based resilience intervention had no effect on resilience or rates of mental ill health.[4] More effective preventative interventions for emergency workers are urgently needed.

Established interventions may have been unsuccessful because they fail to target predictors of mental ill health and are offered to emergency workers after rather than before repeated exposure to the stresses linked to their work. Moreover, cognitive strategies that could help them cope with characteristic stressors are not included as part of the training. For example, our and others' research has demonstrated that exposure to trauma or stressful scenarios through imagery reduces anxiety for police officers and other at risk populations.[5 6] Development of more effective interventions requires identification of predictors of mental disorders and an understanding of how to modify them.

In a series of experimental and prospective studies, we identified two cognitive factors that are robust predictors of poor mental health in emergency workers: rumination (repetitive negative thinking) and resilience appraisals. Those who reported ruminative thoughts during critical incidents were more likely to experience poor levels of coping.[7] Adaptive appraisals during analogue trauma led to more successful attempts to regulate emotions and fewer PTSD symptoms.[8] Our large-scale prospective study of newly recruited paramedics investigated predictors of PTSD and MD derived from cognitive theories of PTSD and depression.[1] Rumination at the start of paramedic training uniquely predicted PTSD; low resilience uniquely predicted an episode of MD.

We then developed an intervention to modify peritraumatic ruminative thinking (ie, thinking repetitively in an abstract way during trauma). Training to think in a concrete style (eg, focusing on objective details and the sequence of events) led to significantly fewer intrusive memories and PTSD symptoms than individuals trained in a ruminative style.[9] We also applied one of the core techniques of a successful treatment for PTSD (cognitive therapy for PTSD[10]), updating the memory of the stressful event with helpful information, as a preventative strategy for dealing with analogue trauma and found that it is more helpful in reducing repetitive thinking and PTSD symptoms than control interventions including exposure.[11]

Research has further demonstrated that exposure to trauma or stressful scenarios through imagery reduces anxiety for police officers and other at risk populations, and that internet-based cognitive treatment that includes attention training as a core component significantly reduces anxiety.[5 6 12]

### Neurobiological factors linked to PTSD and MD

Genetic and longitudinal studies suggest that inflammation is a pre-existing vulnerability factor for the development of PTSD in trauma-exposed individuals rather than simply a correlate of subjective distress, disease severity or maladaptive coping strategies following PTSD onset.[13 14] For example, brain imaging studies have shown that high inflammation levels may increase threat perception (negative valence). Peripheral administration of lipopolysaccharides residues from bacterial cells' components known to elicit a strong systemic inflammatory response, potentiates amygdala activity in response to socially threatening stimuli (fear faces).[15] In turn, greater pretreatment amygdala reactivity to threat predicts less symptom reduction during cognitive behaviour therapy.[16] Additionally, inflammation is an important risk factor for depression and cardiovascular disease, which frequently accompany PTSD.[17-19] Our study will investigate the link between inflammation and the development of PTSD and MD in trauma-exposed student paramedics. We will investigate whether or not internet cognitive training for resilience (iCT-R) can reduce levels of clinically relevant inflammation levels, such as C-reactive protein (CRP), known to increase risk of psychiatric as well as cardiovascular and metabolic conditions comorbid with PTSD and MD.

Given the wealth of literature supporting a relationship between the stress hormone, cortisol and PTSD and MD, we will also systematically assess the cortisol awakening response (CAR) and diurnal cycle. The CAR is an endocrine marker, defined as the change in cortisol concentration that occurs during the first hour after waking from sleep.[20] A meta-analysis of 62 studies concluded that increases in the CAR were associated with job stress and life stress and linked to greater fatigue, burnout and exhaustion and risk for later health states, such as coronary heart disease.[21] A recent study found that higher CAR predicted future episodes of MD within a 2.5-year period.[20] We anticipate that iCT-R will reduce the CAR and cortisol throughout the day and protect against the development of PTSD and MD.

### Study objectives

The primary aim of the study is to evaluate the efficacy of iCT-R. We hypothesise that iCT-R will lead to fewer cases of PTSD and major depression (including subsyndromal PTSD and MD) and less severe PTSD and MD symptomatology at follow-up compared with an existing online training (Mind-Online) and standard practice.

### Secondary objectives

We hypothesise that iCT-R will lead to greater improvement in secondary outcome measures (resilience, rumination, hormone and immune function, smoking, weight gain, alcohol use, symptoms of anxiety and sleep problems, psychological distress, well-being) than Mind-Online and standard practice. We also expect that iCT-R will be more cost-effective than Mind-Online and standard practice because of lower cost per participant without an episode or with low symptoms of PTSD or MD and lower costs per quality adjusted life years (QALY) gained for participants receiving iCT-R.

## Tertiary objectives

We want to establish which baseline factors influence the effect of the interventions on primary and secondary outcomes so that we may make inferences about mechanisms of intervention efficacy. Understanding the effects that modifying risk and protective factors have may drive the refinement of future interventions. Our tertiary objectives are to determine which psychiatric, personality, trauma and social support factors at baseline (social support, trauma exposure, anxiety, age, gender, education, neuroticism, past and current psychiatric status, immune function) may influence (ie, moderate) the effect of the interventions on levels of symptoms (PTSD or MD), psychological distress and well-being at follow-up. Determining which factors moderate outcome may inform improvements to the intervention. For example, should baseline factors, such as education or age moderate outcome, then the intervention could be improved in light of relevant moderators. This could include making it more accessible to younger participants with less education should this be relevant, for example. We will also investigate whether or not changes in resilience-related factors (rumination, responses to intrusions, concrete thinking, resilience appraisals, practice of iCT-R/Mind-Online tools) mediate symptom levels of PTSD and MD at 1-year and 2-year follow-up with iCT-R and Mind-Online. Finally, we will investigate whether or not concrete thinking, practice of tools and responses to intrusions at 6 months predict diagnoses and levels of PTSD and depression symptoms at 1-year and 2-year follow-up.

## Methods

The protocol includes all details required for the WHO Trial Registration Data Set (online appendix 1) and was written in line with the Standard Protocol Items: Recommendations for Interventional Trials statement, which outlines recommendations for a minimum set of scientific, ethical and administrative elements that should be addressed in a clinical trial (online appendix 2).[22]

## Design

The proposed study is a single-blind (assessors blinded) randomised controlled trial in which n=570 student paramedics will be randomly allocated to receive iCT-R, an already available intervention (Mind-Online) that has been investigated in previous trials or standard practice. Participants are also invited to give salivary and plasma samples before and after the interventions and at 1-year and 2-year follow-up. The trial will take place from October 2017 to January 2021.

## Participants

Student paramedics will be recruited from collaborating paramedic training programmes (University of Brighton, Oxford Brookes University, Bournemouth University, University of Hertfordshire, University of Worcester, University of Surrey and Anglia Ruskin University). The locations selected constitute rural and city locations to improve generalisability. The researchers will present the study to each year group at collaborating universities to ensure the maximum reach of recruitment. After presenting the study, researchers will collect names and email addresses of interested students and email the registration survey including the participant information sheet (online appendix 3).

## Inclusion and exclusion criteria

Students who are aged 18 and above, are training to be paramedics and are in years 1, 2 or 3 of their paramedic training programme will be eligible for the study. They will be screened for levels of PTSD and MD, and a trained research assistant will contact participants if they score in the clinical range on measures of PTSD or MD, or report suicidal ideation to evaluate whether they are eligible or need treatment (under JW's supervision). The screening survey will trigger automatic notifications to the research assistant and the principal investigator if a participant scores 10 or above on the Patient Health Questionnaire 9 (PHQ-9)[23] or 1 or above on the suicidal ideation item of the same questionnaire or 33 or above on the Post-traumatic Stress Disorder Checklist for Diagnostic and Statistical Manual of Mental Disorders, fifth edition (PCL-5).[24] Participants will be excluded from the study if their symptoms are interfering with their lives and they would like treatment, and the research assistant will offer them information on how to access evidence-based treatment for these conditions in local services.

## Sample size calculation

The risk of student paramedics developing full syndromal PTSD and MD over 2 years without intervention is 10%, and 25% if subsyndromal PTSD and MD are included.[1] Since there are no interventions for emergency workers which target modifiable risk factors, we referred to a study with a similar approach to facilitate the calculation of power. Topper *et al* evaluated the effects of an intervention targeting rumination on rates of depression in adolescents at 1-year follow-up in comparison to a waitlist condition.[25] The intervention reduced the rates of depression by 67% in comparison to the wait list condition. We estimated that our intervention, which also aims to modify rumination, would reduce rates of PTSD and depression by 50% in comparison to an existing intervention which has shown no change in rates of PTSD or MD over time.[4] Setting power at 80%, α=0.05 and hypothesising a reduction of relative risk of 50% gives an OR of 0.429, which requires a total sample size of n=304 to show a risk reduction of 50% between iCT-R and the alternative intervention. Thus, each condition would require n=152. Since we have a third condition (standard practice), the total sample size required would be n=456. Allowing for a 20% rate of attrition, we will require a total sample size of n=570.

## Randomisation and blinding

Participants will be randomised on a 1:1:1 ratio as per a computer-generated randomisation schedule stratified

by site, gender and baseline PHQ-9 score (≥9 vs <9) and PCL-5 score (≥33 vs <33). The Oxford Clinical Trials Research Unit is independent to the research team and developed the randomisation programme. The researchers will inform participants of the intervention they are to receive after they have completed baseline assessments. Outcome assessment will be single blind; questionnaires are completed online without any involvement of the researchers, the clinical interview will be conducted by an independent assessor blind to treatment allocation, and all personnel involved in processing and assessing the blood and saliva samples will be blinded to treatment allocation. Due to the nature of the interventions, participants cannot be completely blinded to allocation.

However, the inclusion of an already available, alternative intervention aims to mitigate some risk of bias.

### Intervention arms
iCT-R aims to modify rumination and appraisals linked to low resilience in a six-session supported online intervention. We include an imagery component, practice of strategies that has been shown to prevent stress-related responses from developing,[8 9] attention training[12] and monthly top-up exercises during follow-up to consolidate training, an approach that is lacking with existing interventions.

Our intervention follows a similar format to the internet-based programmes that Clark, Ehlers, Wild and colleagues have developed for social anxiety and PTSD.[12 26] The core information is delivered in six modules. The modules include whiteboard videos to explain concepts, audio files for practising concrete thinking, testimonies from qualified paramedics and video footage of student paramedic call-outs for use in experiential exercises. Following our findings of the protective benefits of concrete thinking[8] and the wealth of work in this area,(ie,[27]) participants are regularly reminded to practise concrete thinking.

The modules are:
1. It Matters What you Focus On: Helpful and Unhelpful Attention.
2. Get Out of Your Head with Helpful Thinking.
3. Habits and Dwelling: How to Change Them.
4. Dealing with Unwanted Memories: Then versus Now.
5. Transforming Worries and Improving Performance.
6. Beating Stress and Trauma: My Blueprint.

A trained online coach (research assistant) provides email feedback on students' responses and, through an automated SMS programme, sends regular brief reminders of key points and notifications to practice IF-THEN plans (a technique shown to help individuals respond to warning signs for stress and dwelling).

### Mind-Online
The alternative intervention is a series of six modules available online covering information and advice about stress, sleep problems, anger, depression, PTSD and mindfulness. Participants will receive the same frequency, type and duration of remote support as in iCT-R.

### Standard practice
The third condition is training as usual. Participants will have access to the usual support offered through their university but they will not receive any online modules or remote support. They will be offered the iCT-R at the end of follow-up, when the study is completed.

### Primary outcome measures
#### Levels of PTSD and MD
An independent assessor will administer the PTSD and MD modules of the Structured Clinical Interview for DSM-5 (SCID-5) to assess clinical and subsyndromal PTSD and MD.[28] The SCID-5 is an interview schedule for determining DSM-5 psychiatric diagnoses. PTSD and MD symptomatology will also be assessed with continuous measures: the PCL-5 and the PHQ-9,[23 24] which will be completed at screening, which is typically the same day or shortly before the baseline questionnaires are released and completed. The PHQ-9 and PCL-5 scores at screening will be used as baseline scores in analyses. The PCL-5 is a self-report measure consisting of 20 questions that parallel the diagnostic criteria for PTSD set out in the Diagnostic and Statistical Manual of Mental Disorders, fifth edition.[29]

Items are scored on a five-point Likert scale (0='not at all' to 4='extremely'). The PHQ-9 is a well-validated nine-item self-report measure that assesses symptoms of depression. Items are scored on a four-point Likert scale (0='not at all' to 3='nearly every day'). See table 1 for the full list of outcomes, measures and assessment time points.

### Secondary outcomes
#### Psychological outcomes
Two measures of resilience will be administered: the Wagnild Resilience Scale and the Connor-Davidson Resilience Questionnaire (CD-RISC).[30 31] Two measures of resilience will be used since it is unclear which one most sensitively measures resilience in student paramedics. The Wagnild Resilience Scale is a 25-item scale that measures resilience by rating responses to statements on a seven-point Likert scale (0='strongly disagree' to 7='strongly agree'). The CD-RISC is a commonly used self-report measure of resilience with 25 items scored on a five-point Likert scale (0 = 'not true at all' to 4 = 'true nearly all the time'). Both scales are well-validated measures with excellent psychometric properties. Rumination will be assessed with the brooding subscale of the Ruminative Responses Scale, a reliable and valid measure of the frequency of engaging in dwelling.[32] Rumination in response to unwanted memories will be assessed with the dwelling subscale of the Responses to Intrusions Questionnaire, a reliable and valid measure of maladaptive responses to intrusive memories.[33] Anxiety will be measured with the Generalised Anxiety Disorder scale 7, a seven-item scale with items scored on a four-point Likert scale (0='not at

**Table 1** Outcomes and measures

| Domain | Measures | Time point* |
|---|---|---|
| **Primary outcomes** | | |
| PTSD | Structured Clinical Interview for Diagnostic and Statistical Manual of Mental Disorders, fifth edition Disorders (SCID-5)[28] Post-traumatic Stress Disorder Checklist for DSM-5[24] | 0 1 3 4<br>0 1 2 3 4 |
| MD | SCID-5.[28] Patient Health Questionnaire 9[23] | 0 1 3 4<br>0 1 2 3 4 |
| **Secondary outcomes** | | |
| Resilience | Connor-Davidson Resilience Questionnaire[31]<br>Wagnild Resilience Scale[30] | 0 1 3 4<br>0 1 3 4 |
| Rumination | Ruminative Responses Scale brooding subscale[32]<br>Responses to Intrusions Questionnaire dwelling subscale[33] | 0 1 3 4 |
| Anxiety | Generalised Anxiety Disorder seven-item scale[34] | 0 1 3 4 |
| Smoking and Alcohol use | Smoking Behaviour Questionnaire.[1] Alcohol Use Questionnaire[1] | 0 1 3 4 |
| Weight and Height | Questionnaire recording participants' height and weight[1] | 0 1 3 4 |
| Psychological distress | General Health Questionnaire[35] | 0 1 3 4 |
| Well-being | Warwick Edinburgh Mental Well-being Scale[36] | 0 1 3 4 |
| Hormone function | Level of cortisol in response to awakening and throughout the day | 0 1 3 4 |
| Immune function | Level of C-reactive protein | 0 1 3 4 |
| Sleep problems | Insomnia Severity Index[37] | 0 1 3 4 |
| Health economics | EuroQol 5 Dimensions Questionnaire[38] | 0 1 3 4 |
| | Trimbos/iMTA Questionnaire for Costs Associated with Psychiatric Illness[41] | 0 1 3 4 |
| | Client Service Receipt Inventory[39] | 0 1 3 4 |
| | Health and Labour Questionnaire[40] | 0 1 3 4 |
| **Tertiary outcomes** | | |
| Neuroticism | Eysenck Personality Questionnaire neuroticism subscale[42] | 0 1 3 4 |
| Social support | Social Support scale adapted from a brief measure of social support[43] | 0 1 3 4 |
| Demographics | General information questionnaire[1] | 0 3 4 |
| Trauma exposure | Trauma screener[44] | 0 1 2 3 4 |
| Concrete thinking | Concrete thinking questionnaire, adapted from a previous concrete thinking assessment[43] | 0 1 2 3 4 |
| Intrusions | Duration, frequency and distress linked to Intrusions Questionnaire[44] | 0 1 2 3 4 |

*Time point: 0=baseline, 1=postintervention, 2=6-month follow-up, 3=12-month follow-up, 4=24-month follow-up.

all' to 3= nearly every day').[34] Psychological distress will be measured with the reliable and valid 12-item General Health Questionnaire 12.[35] Well-being will be assessed using the Warwick Edinburgh Mental Well-being Scale (WEMWBS).[36] The WEMWBS has 14 items and is scored on a five-point Likert scale (1='none of the time' to 5='all of the time').

### Hormone and immune function
Salivary cortisol will be assessed by radioimmunoassay (RIA) analysis. A sample of saliva will be collected on awakening, 15, 30 and 60 min after awakening and at 12:00 and 20:00. Baseline high-sensitive CRP plasma levels will be measured using an ILab 600 spectrophotometric method in serum samples.

### Health outcomes
Smoking and alcohol use will be measured with unpublished questionnaires since two of our assessment points (1-year and 2-year follow-up) require participants to report current use as well as changes in alcohol use and smoking over the previous year, a time period not currently referenced in validated tools. Shorter questionnaires may reduce response burden and improve questionnaire completion. Participants will be asked to indicate whether or not they smoke, how many cigarettes they smoke a day and whether this has increased, decreased or stayed the same in the last year. They will also be asked how many units of alcohol they have had in the last week, whether this is an average amount for them, and if not, how many units they usually drink per

week. Weight gain will be measured by increases in body mass index. Participants will be asked to provide their weight and height. The researchers will take weighing scales and a tape measure to study visits to weigh and measure participants. Sleep problems will be assessed by the Insomnia Severity Index, which is a reliable and valid brief self-report instrument of sleep quality and sleep difficulties.[37] In line with National Institute for Health and Care Excellence guidelines, health related quality of life will be measured by the five-levels version of the EuroQol 5 Dimensions questionnaire (EQ-5D-5L).[38] The EQ-5D-5L is a validated and widely used generic measurement of health related quality of life based on five dimensions: mobility, self-care, usual activities, pain/discomfort, and anxiety/depression. Although, we do not expect that all five dimensions will be affected by the intervention, we need to collect data on all dimensions in order to determine quality adjusted life years (QALYs).

## Costs

The economic evaluation will be conducted by taking the National Health Service (NHS) perspective (including the costs of using mental health services) and the broader societal perspective (including the costs of productivity loss due to illness). Mental health resource utilisation will be measured with an adapted version of the Clinical Service Receipt Inventory and valued by using published unit costs (eg, NHS Reference Costs and Unit Costs of Health and Social Care).[39] Productivity loss will be measured with Short Form-Health and Labour Questionnaire,[40] a validated questionnaire part of the Trimbos/iMTA questionnaire for Costs associated with Psychiatric Illness.[41]

## Tertiary outcomes

To assess potential moderators of outcomes, we will measure psychiatric, personality, trauma and social support factors at baseline (social support, trauma exposure, anxiety, age, gender, education, neuroticism, past and current psychiatric status, immune function). The neuroticism subscale (12 items) of the Eysenck Personality Questionnaire has excellent psychometric properties and is a measure of emotionality.[42] We will use an adapted version of a brief measure of social support, to assess perceived support from and closeness to friends, family and work colleagues, as well as use of social support.[43] Trauma exposure will be measured using a 19-item unpublished trauma questionnaire relevant to emergency workers, which includes items from the Life Events Checklist.[44] We will also collect demographic information (age, gender, and level of education), information on the duration, frequency and distress linked to the Intrusions Questionnaire,[45] and questions about concrete and abstract thinking based on an existing assessment tool.[46] Participants will be asked to think about a problem they are having and write questions that may go through their minds in relation to the problem. They will then be presented with four problem scenarios and asked to select

from a list the likely thoughts they would have if faced with the problem. The list consists of a range of concrete and abstract thoughts. We will investigate whether or not changes in resilience-related factors (rumination, responses to intrusions, concrete thinking, resilience appraisals, practice of iCT-R/Mind-Online tools) mediate symptom levels of PTSD and MD at 1-year and 2-year follow-up with iCT-R and Mind-Online. We will also investigate whether or not concrete thinking, practice of tools and responses to intrusions at 6 months predict diagnoses and levels of PTSD and depression symptoms at 1-year and 2-year follow-up.

## Patient and public involvement

We held three User Advisory Groups with student paramedics who contributed to the design of the study, the selection of questionnaires and the content of the intervention. The first User Advisory Group was co-organised with the Research Design Service South Central Patient and Public Involvement Officer, Megan Barlow Pay. Research questions and outcome measures were discussed with all feedback incorporated, including the development of a module participants requested to address socially anxious concerns common to their combined role as student and paramedic. A further two User Advisory Groups were held to review the intervention and to develop four versions of a questionnaire to assess concrete and abstract thinking in situations specific to student paramedics. Participants also completed the baseline questionnaires to assess their length of time and to provide feedback on the feasibility of administration. Participants were not involved in the recruitment and conduct of the study. Results will be made available in summary format to all participants by email once the study is completed.

## Procedure

Researchers will present the study at collaborating universities and invite student paramedics to take part. Interested participants will be given a weblink to the study via our software platform, Qualtrics, where they can read and print a PDF copy of the Participant Information Sheet and discuss questions with the research assistant over the telephone. If they decide to take part, they will be emailed a link where they can login, re-read the Participant Information Sheet and complete a consent form (see online appendix 3 for the Participant Information Sheet and Consent Form). Written consent will be requested from participants before blood and saliva samples are taken. It will be made clear that participation is entirely voluntary and that volunteers may withdraw from the study at any point without incurring any negative consequences.

Participants will be recruited over a 12-month period. They will complete online two questionnaires (PCL-5 and PHQ-9) to assess eligibility. If participants score ≥10 on the PHQ-9, ≥33 on the PCL-5 or ≥1 on the PHQ-9 suicidal ideation item, a researcher will telephone the participant to determine symptom severity and whether treatment is necessary. A risk assessment will be conducted

if a participant scores ≥1 on the PHQ-9 suicidal ideation item. If treatment is needed, participants will be signposted to their GP and local psychological services. If participants are eligible and consent to participate in the study, they will be assigned a participant ID number to ensure anonymity is maintained. They will complete a set of questionnaires online, take part in a clinical interview conducted by an independent assessor, and provide blood and saliva samples. The clinical interviews will be recorded on SanDisk MP3 recorders to ensure that the questions asked are standardised across all assessment interviews. Qualified phlebotomists will visit students at their universities to collect blood samples. During this study visit, participants will be provided with equipment to collect their saliva at home on awakening and through the day (six samples in total). Saliva and blood samples will be transported to the University of Surrey for assay

analysis. The blood samples will be centrifuged immediately and only serum will be kept for analysis.

Once baseline assessments are completed, participants will be randomly allocated to iCT-R, Mind-Online or standard practice. When the interventions or standard practice are completed, participants will complete the full battery of assessments again; they will complete the questionnaires online, take part in a clinical interview, and provide blood and saliva samples. Six months later, they will complete online a shorter set of questionnaires. At 12-month and 24-month follow-up, they will complete the full battery of assessments again. See figure 1 for a timeline of the study including the enrolment process, randomisation, interventions and assessments. We are aware that there are various tasks to complete at each assessment point. This may be off-putting to participants and increase the likelihood of drop outs and respondent

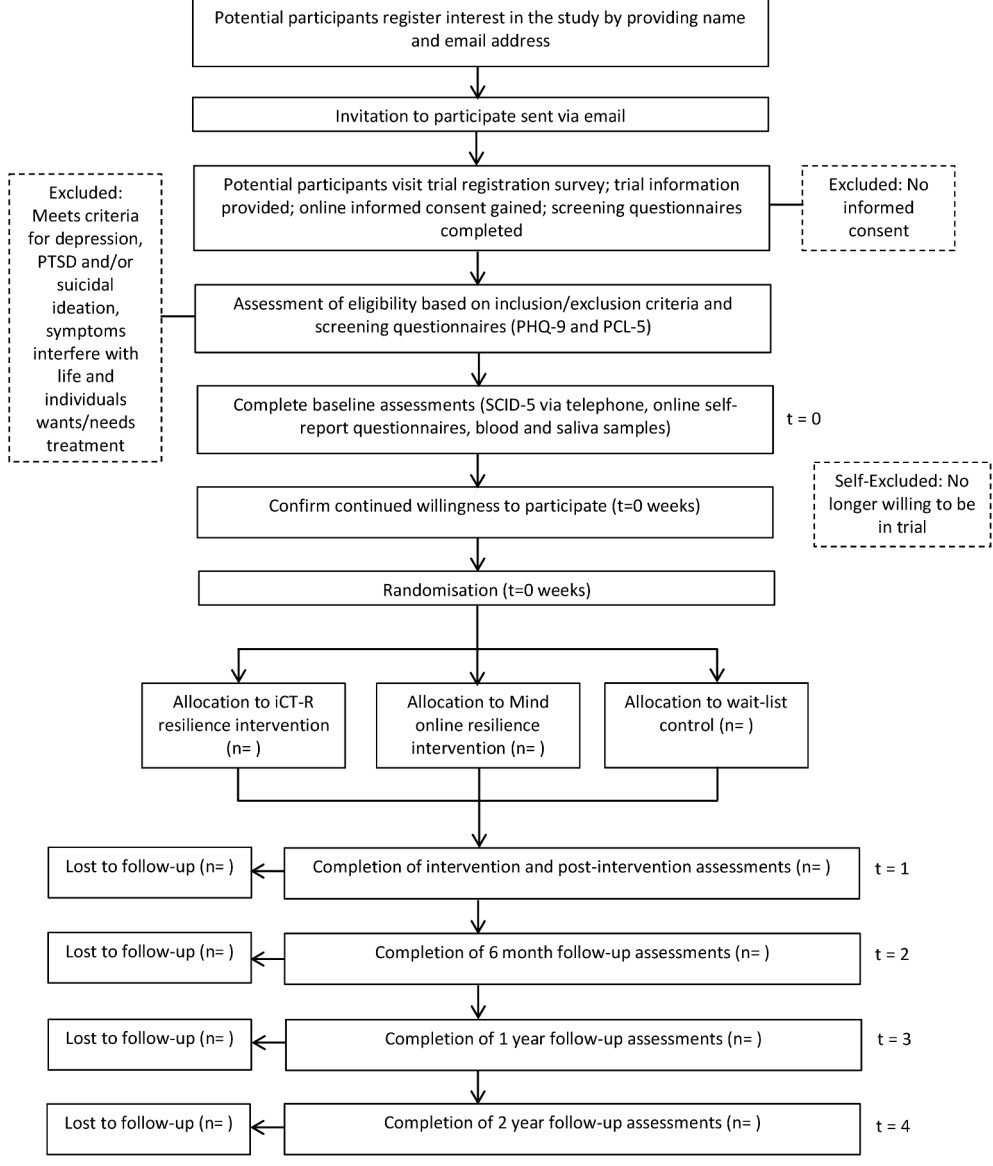

**Figure 1** Study time line. The flow chart shows how participants will progress through the study from the initial stage of enrolment through to analysis.

fatigue. We will clearly communicate the value of the assessments being administered, and participants will be compensated with £30 and a certificate of completion at follow-up time points to discourage drop out. The reasons for non-adherence to the intervention or dropping out of the study will be recorded. An independent rater will assess treatment fidelity. The content of a random sample of email communications will be scored for reference to content relevant to each training programme.

## Data management

Registration and assessment data will be captured online via Qualtrics software. Participants will be assigned a unique code to be used for all data files and audio tapes. Access to the system will be restricted to named study personnel and via password protection. The University of Oxford's IT services have arranged for the files to be encrypted and backed up on a weekly basis using a Tivoli Storage Manager (TSM). The data will be copied to three separate tapes; one copy will reside in the Tape Robot in the IT Services Machine room and the other two are held in locked fireproof safes, one on-site at IT services and one off-site in locked premises. The data on the tapes are inaccessible without the TSM database. The data on the off-site tapes are encrypted. Papers from clinical interviews will be kept in locked cabinets at the University of Oxford. The audiotapes from clinical interviews will be backed up online with password-protection and access restricted to study personnel. The blood samples will be centrifuged as soon as the laboratory at the University of Surrey receives them on the day of collection. The cellular component will be discarded and the serum will be stored at −80°C. Saliva samples will be analysed at the University of Surrey by RIA analysis to detect levels of cortisol.

In line with the Oxford Clinical Trials Research Unit and the Medicines for Human Use Clinical Trials Regulations (2004), we have not recruited a Data Monitoring Committee because recruitment and follow-up occur over a short period, there are minimal risks to participants and the trial protocol will not be modified regardless of the interim data.

## Statistical analyses

In line with the *BMJ* and Consolidated Standards of Reporting Trials guidelines, data analysis will be intent-to-treat. All participants who have been randomised will be included in analyses, including those who drop out. We will compare dichotomous measures (rates of PTSD and MD, changes in alcohol use and smoking) between conditions using $X^2$ analysis. Continuous measures will be analysed using hierarchical linear modelling. This analysis models random slopes and intercepts for participants and tests the fixed effects of repeated assessments over time (level 1, preintervention, postintervention, 1 and 2 year follow-up) and training condition (level 2, iCT-R, Mind-Online, Standard Practice) using data from all participants. It takes into account that participants are nested within site (level 3). Variables will be centred for

the analysis. The effects of potential moderators (social support, exposure to critical incidents, etc) on PTSD and depression symptoms will be explored by including main effects and interactions with treatment effects into the model. Non-significant moderators will be removed from the final model.

To address the potential for Type I error when evaluating our secondary outcomes (ie, resilience, rumination, hormone and immune function, smoking, weight gain, alcohol use, anxiety, sleep problems, psychological distress, well-being), we will examine and report effect sizes. Effect sizes are a reliable method for determining the quality of the result that do not rely on p value significance and are not affected by the number of outcomes.

Mediation analyses will be conducted to assess whether or not changes in resilience-related factors (rumination, responses to intrusions, concrete thinking, resilience appraisals, practice of iCT-R/Mind-Online tools) and compliance with the training programmes mediate symptom levels of PTSD and MD at one and 2 year follow-up with iCT-R and Mind-Online.

A trial-based economic evaluation will be conducted to investigate the cost-effectiveness of the intervention in terms of cost per QALY gained. Uncertainty in the results will be addressed in sensitivity analyses and displayed in cost-effectiveness planes and cost-effectiveness acceptability curves.

## Adverse events

We do not anticipate any adverse events. However, it is possible that a participant may evidence risk at one of the assessment points (preintervention, postintervention or 12 and 24 month follow-up). If this is the case, risk will be assessed over the telephone and the individual will be signposted to the appropriate service.

Should a serious adverse event (SAE) occur where, in the opinion of the principal investigator, the event was 'related' (resulted from administration of any of the research procedures) and 'unexpected' in relation to those procedures, it will be reported to the research ethics committee. Reports of related and unexpected SAEs will be submitted within 15 working days of the principal investigator becoming aware of the event, using the Health Research Authority safety report form for a non-Clinical Trial of an Investigation of a Medicinal Product (non-CTIMP).

The University of Oxford has a specialist insurance policy in place, which would operate in the event of any participant suffering harm as a result of their involvement in the research (Newline Underwriting Management Ltd, at Lloyd's of London).

## Ethics and dissemination

Ethical approval of the research protocol was gained from The Medical Sciences Inter-Divisional Research Ethics Committee at the University of Oxford, 17 August 2017, ref: R44116/RE001. This is protocol version 1. Any substantive amendments to the protocol will be conducted

by the principal investigator and reviewed by the research ethics committee.

The research results will be submitted for publication in a peer-reviewed journal and presented at relevant conferences. Direct access to data will be granted to authorised representatives from the host institution and the regulatory authorities to permit trial-related monitoring, audits and inspections.

## Committees

We have established a trial oversight committee (TOC). Our independent chairman is Dr Susan Dutton, the Senior Medical Statistician and Oxford Clinical Trials Research Unit Lead Statistician. The principal investigator is also a member of the TOC and we have one lay qualified paramedic member (Graham Harris). The TOC will meet before the start of the trial and three more times before the end of the trial.

**Author affiliations**
[1]Department of Experimental Psychology, Oxford Centre for Anxiety Disorders and Trauma, University of Oxford, Oxford, UK
[2]Oxford Health NHS Foundation Trust, Oxford, UK
[3]Department of Psychological Medicine, Institute of Psychiatry, Psychology and Neuroscience, King's College London, London, UK
[4]Department of Social, Genetic and Developmental Psychiatry, Institute of Psychiatry, Psychology and Neuroscience, King's College London, London, UK
[5]Department of Child and Adolescent Psychiatry, Institute of Psychiatry, Psychology and Neuroscience, King's College London, London, UK
[6]Nuffield Department of Population Health, Health Economics Research Centre, University of Oxford, Oxford, UK
[7]Sir Henry Wellcome Building for Mood Disorders Research, School of Psychology College of Life and Environmental Sciences, University of Exeter, Exeter, UK
[8]Faculty of Health and Medical Sciences, University of Surrey, Surrey, UK
[9]School of Health Sciences, University of Brighton, Brighton, UK

**Acknowledgements** We are grateful to Dr Esther Beierl for conducting the independent statistical analyses for this study and to Dr Susan Dutton for chairing the Trial Oversight Committee. We are grateful to Megan Barlow Pay for facilitating the first User Advisory Group. We thank our User Advisory Group members who have contributed to the development of the intervention and the design of the study.

**Contributors** All authors contributed substantially to conception and the design of the protocol or the acquisition of data for the work. All revised the manuscript critically for important intellectual content, all approved the final manuscript, and all are accountable for all aspects of the work in ensuring that questions related to the accuracy or integrity of any part of the work are appropriately investigated and resolved. JW is the principal investigator and oversaw the study and arranged and lead meetings with a User Advisory Group and the Trial Oversight Committee. JW conceived and co-designed the study with AE. JW developed the intervention with AE, EW and GT. JW completed the ethics application and liaised with collaborators to support recruitment, managed and supervised the research team and wrote the primary paper for publication. AB facilitated recruitment and contacted with collaborating centres. GT offered online support to participants receiving the online interventions and helped to schedule assessments. SE is the independent assessor at baseline and follow-up. JW, GT, SE and HL are responsible for data collection. BM analysed the biological data. CP, AD and AT guided the analyses of the biological and health economics data.

**Funding** This work is funded by an MQ: Transforming Mental Health grant (number CQR01260) and supported by the NIHR Oxford Health Biomedical Research Centre. MQ had no role in the design of this study and will not have any role during its execution, analyses, interpretation of the data or decision to submit results. AE is funded by a Wellcome Trust Principal Research Fellowship (grant 200796). CP is supported by the NIHR Biomedical Research Centre at the South London and Maudsley NHS Trust and King's College London, London, UK.

**Disclaimer** The views expressed are those of the authors and not necessarily those of the NHS, the NIHR or the Department of Health.

**Competing interests** JW, AE and their team have developed iCT-R. They do not receive any income from this work.

**Patient consent for publication** Not required.

**Ethics approval** The Medical Sciences Inter-Divisional Research Ethics Committee at the University of Oxford.

**Provenance and peer review** Not commissioned; externally peer reviewed.

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
