## [Reviewer comments · BMJ Open]

This paper was submitted to a another journal from BMJ but declined for publication following peer review. The authors addressed the reviewers' comments and submitted the revised paper to BMJ Open. The paper was subsequently accepted for publication at BMJ Open.

(This paper received three reviews from its previous journal but only two reviewers agreed to published their review.)

ARTICLE DETAILS

TITLE (PROVISIONAL)	Preventing PTSD, depression, and associated health problems in student paramedics: Protocol for PREVENT-PTSD, a randomised controlled trial of supported online cognitive training for resilience versus alternative online training and standard practice
AUTHORS	Wild, Jennifer; El-Salahi, Shama; Tyson, Gabriella; Lorenz, Hjördis; Pariante, Carmine; Danese, Andrea; Tsiachristas, Apostolos; Watkins, Edward; Middleton, Benita; Blaber, Amanda; Ehlers, Anke

VERSION 1 – REVIEW

REVIEWER	Herta Flor, PhD Central Institute of Mental Health Medical Faculty Mannheim, Heidelberg University Mannheim Germany
REVIEW RETURNED	09-Mar-2018

GENERAL COMMENTS	This is an excellent description of a planned trial. The intended intervention is state of the art . I only have two suggestions: 1) Would it be possible to use drug screening (for smoking, alcohol, other drugs) rather than just questionnaires? 2) Is the credibility of the treatment assessed at the beginning and end of the trial to ensure comparability of the experimental and control intervention?
--

REVIEWER	David McBride Department of Preventive and Social Medicine Dunedin School of Medicine Dunedin New Zealand
REVIEW RETURNED	05-Apr-2018

GENERAL COMMENTS	This is a very interesting and somewhat challenging research proposal thank you! My major problem is with the multiple outcome measures, firstly in the load that it would place on volunteers and secondly on the interpretation of the results. I do note that you have ethics approval. The aims are clear, reducing PTSD and MD in this population would help reduce morbidity from psychological disorders, and perhaps future physical health. Rumination and resilience seems to be important modifiers, and imagery a good idea. Page 7, the introduction of the inflammatory concept into the scheme of things introduces a level of complexity into what is, up to
---

now, a clear concept.

The primary objective is clear, fewer cases of PTSD and MD. We then have primary, secondary and tertiary objectives.

Some of the secondary objectives are also conceptually intuitive, in that resilience (within the constraints of your construct) should be increased and rumination reduced. I can also see the point in looking at weight gain alcohol use and smoking, however:

1. anxiety and psychological distress also seem to be primary outcomes (we find out much later about the 'clinical screening.')

We also have QALYs, in the form of the EUROQoL, in which mental health is being measured so that the economic evaluation can be carried out.

2. The action on cortisol and inflammatory markers would be 'nice to know,' but are they an essential part of the PTSD and MD reduction hypothesis?

The tertiary objectives seem to be taking us into different territory altogether, they introduce new factors, which although making conceptual sense, come somewhat 'out of the blue.'

3. I can see that it would be very useful to see how risk and protective factors moderate PTSD and MD, however would the practical application of the intervention require measurement of these factors at recruitment? You need to explain why, and how, this knowledge would help the efficacy of the intervention, if indeed it will.

The methods are clear.

4. Page 14 line 17. The SCID-5, using as it does a structured clinical interview, would seem to be the gold standard for assessing PTSD and MD.

5. The PCL-5 and PHQ 9 give ordinal scales and could be used for the same purpose, but you also seem page 20, line 28, to use it in screening. Which are you going to use in the analysis?

6. I think also that you need to justify the use of any additional measures in the design: they increase the burden on the volunteers.

7. More so with two measures of resilience! Then we have the rest of the instruments. Could you not consolidate some of these constructs? Why, for example, do you need to use both the GHQ 12 and the WEMWBS?

8. When and how often are cortisol samples to be taken?

9. Page 16 first line. Any use of unpublished instruments for smoking and alcohol will not facilitate replication. You do explain the questions, but also need to state if and how they are to be incorporated in the analysis.

10. The process analysis section relates to the tertiary objectives, and yes, you should have referred to table 1 at a much earlier point in the discourse.

The procedures are clear.

11. Page 20 line 6, you now make it clear that the interview is a risk assessment. Is this part of the normal recruit procedure?

12. Why not use a suitable cut-off on the PCL-5 instead?

Analyses

13. Having included multiple outcomes, you must be very clear at this point how you are going to carry out the proposed modelling,

	how the variables are going to be included/excluded and in particular how you are going to adjust for the multiple outcomes. 14. You also need to say something about participant load and the effect that it might have on introducing the intervention 'in practice.'
--	--

REVIEWER	Mark Brown, Ph.D. Associate Professor Bradley University Foster College of Business Administration Department of Management and Leadership Peoria, Illinois USA 61606
REVIEW RETURNED	13-Apr-2018

GENERAL COMMENTS	This is a nicely designed study protocol for investigating the efficacy of a newly developed intervention to prevent PTSD and associated health issues among student paramedics. I have but one minor issue with the protocol and it involves the measurement of smoking and alcohol use in the context of health outcomes. As we are all aware, self reported data can be inaccurate and this is particularly the case with socially stigmatized behaviors such as cigarette smoking and alcohol consumption. However, I am more than confident the investigators are well aware of this, and seeing as to the fact there is no alternative way to assess these I view this as only a minor concern. I thank the authors for their efforts on behalf of the health and well being of health care professionals and wish them good luck in their future research efforts.
---

VERSION 1 – AUTHOR RESPONSE

Reviewer #1:

1. Would it be possible to use drug screening (for smoking, alcohol, other drugs) rather than just questionnaires?

Thank you for this suggestion. We are aware that using self-report measures for smoking, alcohol and other drug use is less reliable than drug screening. Although we would have liked to have included drug screening to improve reliability, it was not possible due to the costs involved. However, this may be something to consider in future trials.

2. Is the credibility of the treatment assessed at the beginning and end of the trial to ensure comparability of the experimental and control intervention?

The two interventions consist of six modules each, completed on a weekly basis over six weeks. At the end of each module, participants rate how helpful (out of 100%) they found the module. At the end of the course, participants rate how helpful they found the course overall. Helpfulness ratings are being used as a proxy assessment of credibility and will be analysed to determine comparability of the interventions.

Reviewer #2

1. The action on cortisol and inflammatory markers would be 'nice to know,' but are they an essential part of the PTSD an MD reduction hypothesis?

Thank you for your valuable question. Previous research has demonstrated the effects of inflammation and cortisol as pre-existing vulnerability factors to the development of PTSD and MD (Michopoulos et al., 2015; Eraly et al., 2014; Vrshek-Schallhorn et al., 2013). We are interested in whether our resilience intervention can modify risk factors and in turn reduce the likelihood of participants developing PTSD and MD. We thought it would be important to measure the effects on cortisol and inflammatory markers since such markers are vulnerability factors to developing PTSD

and MD and changes in them may be associated with the interventions and may even mediate outcome.

3. The tertiary objectives seem to be taking us into different territory altogether, they introduce new factors, which although making conceptual sense, come somewhat 'out of the blue.' I can see that it would be very useful to see how risk and protective factors moderate PTSD and MD, however would the practical application of the intervention require measurement of these factors at recruitment? You need to explain why, and how, this knowledge would help the efficacy of the intervention, if indeed it will.

Thank you for the suggestion. We have made the following clarification in the Tertiary Objectives section on page 8:

Our tertiary objectives are to determine which psychiatric, personality, trauma and social support factors at baseline (social support, trauma exposure, anxiety, age, gender, education, neuroticism, past and current psychiatric status, immune function) moderate the effect of the interventions on levels of symptoms (PTSD or MD), psychological distress and wellbeing at follow-up. Determining which factors moderate outcome may inform improvements to the intervention. For example, should baseline factors, such as education, gender, age and so on moderate outcome, then the intervention could be improved in light of relevant moderators. This could include making it more accessible to younger participants with less education should this be relevant, for example.

4. Page 14 line 17. The SCID-5, using as it does a structured clinical interview, would seem to be the gold standard for assessing PTSD and MD.

Thank you for highlighting this point. We use the SCID-5 at all core assessment points: baseline, post-intervention and at 1 and 2 year follow-up.

5. The PCL-5 and PHQ 9 give ordinal scales and could be used for the same purpose, but you also seem page 20, line 28, to use it in screening. Which are you going to use in the analysis?

The PCL-5 and PHQ-9 are completed only once at screening which is typically the same day they are given the baseline questionnaires for completion. At screening, if eligible, participants are immediately given the link to the baseline questionnaires. The PCL-5 and PHQ-9 have been omitted from the baseline questionnaire pack so as to avoid repetition and to reduce the load put on participants.

We have made the following changes to the manuscript:

On page 13, in the Primary Outcome Measures:

PTSD and MD symptomatology will also be assessed with continuous measures: the PCL-5 and the PHQ-9 [23, 24], which will be completed at screening, which is typically the same day or shortly before the baseline questionnaires are released and completed. The PHQ-9 and PCL-5 scores at screening will be used as baseline scores in analyses.

6. I think also that you need to justify the use of any additional measures in the design: they increase the burden on the volunteers.

We are conscious that participants are being asked to complete numerous measures. We have taken steps to reduce this load, such as removing the PHQ-9 and PCL-5 from the baseline questionnaire pack as they will have already been completed at screening. We also use very brief measures of alcohol and smoking behaviour. We take on board the need to justify the use of additional measures.

We have added the following to the manuscript:

On page 14-15 in Psychological outcomes:

Two measures of resilience will be administered: the Wagnild Resilience Scale and the Connor-Davidson Resilience Questionnaire (CD-RISC). [30, 31] Two measures of resilience will be used since it is unclear which one most sensitively measures resilience in student paramedics.

On page 17 in Health outcomes:

Smoking and alcohol use will be measured with unpublished questionnaires since two of our assessment points (1 and 2 year follow-up) require participants to report current use as well as changes in alcohol use and smoking over the previous year, a time period not currently referenced in validated tools. Shorter questionnaires may reduce response burden and improve questionnaire completion.

7. More so with two measures of resilience! Then we have the rest of the instruments. Could you not consolidate some of these constructs? Why, for example, do you need to use both the GHQ 12 and the WEMWBS?

We administer the GHQ-12 (the briefest version possible of the GHQ-28) which captures psychological distress. Whilst we are aware that the GHQ-12 and WEMWBS, which measures wellbeing, are correlated, they do measure different constructs and we are interested in capturing both, which will also facilitate comparison with previously published research. We are aware there are many questionnaires and have taken steps to reduce burden on volunteers, such as removing screening questionnaires from the baseline assessment pack and using short measures to assess alcohol and smoking behaviour.

8. When and how often are cortisol samples to be taken?

Cortisol samples will be taken at baseline, post-intervention (approximately 8 weeks later) and at one and two year follow-up. At each assessment point, participants will be provided with the necessary equipment to take six samples of saliva in one day so that we can measure their cortisol awakening response. This information is provided in Table 1 and in the section titled 'Hormone and immune function' on page 15-16.

9. Page 16 first line. Any use of unpublished instruments for smoking and alcohol will not facilitate replication. You do explain the questions, but also need to state if and how they are to be incorporated in the analysis.

Please see page 23-23. We have added the following:

In line with the BMJ and Consort guidelines, data analysis will be intent-to-treat. All participants who have been randomised will be included in analyses, including those who drop out. We will compare dichotomous measures (rates of PTSD and MD, changes in alcohol use and smoking) between conditions using Chi square analysis. Continuous measures will be analysed using hierarchical linear modelling. This analysis models random slopes and intercepts for participants, and tests the fixed effects of repeated assessments over time (level 1, pre-intervention, post-intervention, 1 and 2 year follow-up) and training condition (level 2, iCT-R, Mind-Online, Standard Practice) using data from all participants. It takes into account that participants are nested within site (level 3). Variables will be centred for the analysis. The effects of potential moderators (social support, exposure to critical incidents, etc.) on PTSD and depression symptoms will be explored by including main effects and interactions with treatment effects into the model. Non-significant moderators will be removed from the final model.

10. The process analysis section relates to the tertiary objectives, and yes, you should have referred to table 1 at a much earlier point in the discourse.

Thank you for this suggestion. We have changed the title from 'Process Analyses' to 'Tertiary Outcomes.' For clarity, and we have moved Table 1 to page 14 so that it is earlier in the manuscript.

11. Page 20 line 6, you now make it clear that the interview is a risk assessment. Is this part of the normal recruit procedure?

A risk assessment is not part of the normal recruitment procedure. It will only be conducted if a participant scores 1 or more on the suicidal ideation question of the PHQ-9. If at screening a participant indicates they have suicidal thoughts, a research assistant will telephone them to

determine severity. If treatment is needed the research assistant will signpost the participant for treatment.

We have made the following changes on page 20 to make this process clearer:

If participants score ≥ 10 on the PHQ-9, ≥ 33 on the PCL-5 or ≥ 1 on the PHQ-9 suicidal ideation item, a researcher will phone the participant to determine symptom severity and whether treatment is necessary. A risk assessment will be conducted if a participant scores ≥ 1 on item 9 of the PHQ-9, which assesses suicidal ideation. If treatment is needed, participants will be signposted to their GP and local psychological services.

12. Why not use a suitable cut-off on the PCL-5 instead?

Thank you for highlighting this error. We are using the recommended cut-off of 33 for the PCL-5. This has been amended on page 19.

13. Having included multiple outcomes, you must be very clear at this point how you are going to carry out the proposed modelling, how the variables are going to be included/excluded and in particular how you are going to adjust for the multiple outcomes.

We have added the following to pages 22-23:

In line with the BMJ and Consort guidelines, data analysis will be intent-to-treat. All participants who have been randomised will be included in analyses, including those who drop out. We will compare dichotomous measures (rates of PTSD and MD, changes in alcohol use and smoking) between conditions using Chi square analysis. Continuous measures will be analysed using hierarchical linear modelling. This analysis models random slopes and intercepts for participants, and tests the fixed effects of repeated assessments over time (level 1, pre-intervention, post-intervention, 1 and 2 year follow-up) and training condition (level 2, iCT-R, Mind-Online, Standard Practice) using data from all participants. It takes into account that participants are nested within site (level 3). Variables will be centred for the analysis. The effects of potential moderators (social support, exposure to critical incidents, etc.) on PTSD and depression symptoms will be explored by including main effects and interactions with treatment effects into the model. Non-significant moderators will be removed from the final model.

To address the potential for Type I error when evaluating our secondary outcomes (i.e., resilience, rumination, hormone and immune function, smoking, weight gain, alcohol use, anxiety, sleep problems, psychological distress, wellbeing), we will examine and report effect sizes. Effect sizes are a reliable method for determining the quality of the result that do not rely on p-value significance and are not affected by the number of outcomes.

14. You also need to say something about participant load and the effect that it might have on introducing the intervention 'in practice.'

This is a valid and important suggestion. The following changes have been made to the manuscript on page 20:

We are aware that there are various tasks to complete at each assessment point. This may be off-putting to participants and increase the likelihood of drop outs and respondent fatigue. We will clearly communicate the value of the assessments being administered, and participations will be compensated with £30 and a certificate of completion at follow up time points to discourage drop out.

Reviewer #3

Minor issue involving the measurement of smoking and alcohol use in the context of health outcomes. As we are all aware, self reported data can be inaccurate and this is particularly the case with socially stigmatized behaviors such as cigarette smoking and alcohol consumption. However, I am more than confident the investigators are well aware of this, and seeing as to the fact there is no alternative way to assess these I view this as only a minor concern.

Thank you for your valuable consideration. We agree with you that participants may under-estimate stigmatising behaviours through self-report. To improve reliability of participant responding, we make it clear at every assessment point that our study is independent from students' universities, that participants' responses are private and confidential and will never be fed back to universities. With these statements, which were approved by the UAG and TSC, we hope to encourage honest responding.

We have amended the strengths and limitations box so as to be transparent about the shortcomings of using self-reported measures to assess alcohol and smoking behaviour on page 3:

Smoking and alcohol use will be measured with unpublished self-report tools.

VERSION 2 – REVIEW

REVIEWER	Herta Flor, PhD Central Institute of Mental Health Department of Cognitive and Clinical Neuroscience Medical Faculty Mannheim, Heidelberg University Germany
REVIEW RETURNED	17-Aug-2018
GENERAL COMMENTS	The authors have adequately addressed all my concerns.
REVIEWER	David McBride Preventive and Social Medicine University of Otago
REVIEW RETURNED	05-Aug-2018
GENERAL COMMENTS	Thank you, I am reassured by the answers.